# The Effect of Meloxicam on Inflammatory Response and Oxidative Stress Induced by *Klebsiella pneumoniae* in Bovine Mammary Epithelial Cells

**DOI:** 10.3390/vetsci11120607

**Published:** 2024-11-29

**Authors:** Kangjun Liu, Shangfei Qiu, Li Fang, Luying Cui, Junsheng Dong, Long Guo, Xia Meng, Jianji Li, Heng Wang

**Affiliations:** 1Jiangsu Co-innovation Center for Prevention and Control of Important Animal Infectious Diseases and Zoonoses, College of Veterinary Medicine, Yangzhou University, Yangzhou 225009, China; kangjunliu@yzu.edu.cn (K.L.); qiushangfei1@163.com (S.Q.); lycui@yzu.edu.cn (L.C.); junsheng@yzu.edu.cn (J.D.); yzdxgl@yzu.edu.cn (L.G.); mengxia@yzu.edu.cn (X.M.); jjli@yzu.edu.cn (J.L.); 2International Research Laboratory of Prevention and Control of Important Animal Infectious Diseases and Zoonotic Diseases of Jiangsu Higher Education Institutions, Yangzhou 225009, China; 3Joint International Research Laboratory of Agriculture and Agri-Product Safety of the Ministry of Education, Yangzhou 225000, China; 4School of Veterinary Medicine, Jiangsu Agri-Animal Husbandry Vocational College, Taizhou 225300, China; 2023010549@jsahvc.edu.cn

**Keywords:** *K. pneumoniae*, meloxicam, infection, anti-inflammatory, antioxidant

## Abstract

Meloxicam in combination with antibiotics therapy has showed promising effects in clinical mastitis. The direct effects of MEL on bovine mammary epithelial cells are still not fully elucidated. Moreover, the effectiveness of meloxicam on the inflammatory response and oxidative stress induced by *K. pneumoniae* are also unclear. This study investigates the effects of meloxicam on bovine mammary epithelial cells in response to *K. pneumoniae* infection, with a focus on inflammatory response and oxidative stress. Primary bovine mammary epithelial cells were infected with *K. pneumoniae* in the presence or absence of plasma maintenance concentration of meloxicam (0.5 and 5 μM). At these concentrations, meloxicam was able to inhibit the mRNA expression of pro-inflammatory genes and relieve oxidative stress through the increased activity of superoxide dismutase and catalase and the level of total antioxidant capacity. The mechanisms by which meloxicam mitigated the inflammatory response and oxidative stress were partially attributed to inhibiting the nuclear transcription factor-kappa B signaling pathway and improving the activation of the nuclear factor erythroid 2-related factors signaling pathway.

## 1. Introduction

Bovine mastitis leads to considerable economic losses in the dairy industry and impacts animal welfare [1]. With the widespread implementation of mastitis control programs, environmental pathogens such as *Klebsiella* spp., *Escherichia coli* (*E*. *coli*), and *streptococci* have become the most common causes of clinical mastitis [2,3]. *K. pneumoniae* is well recognized as a major environmental pathogen [2,4]. Currently, mastitis caused by *K. pneumoniae* is particularly problematic due to limited antimicrobial efficacy, poor bacteriological cure rate, and rapid progression to toxic shock [2,5,6]. In dairy cows, *K. pneumoniae* infection is prone to severe clinical mastitis with a prolonged and massive inflammatory response [7]. Intramammary infection with *K. pneumoniae* could lead to epithelial cell death, which prevents cows from returning to their pre-disease levels of milk production [8].

Giving the pain and inflammation during mastitis, nonsteroidal anti-inflammatory drugs (NSAIDs) are recommended to relieve the local and systemic clinical signs, thereby promoting the recovery of mammary normal physiology function. MEL, an NSAID, has been licensed for treating the pain and inflammation in clinical mastitis [9]. Compared with other nonspecific COX inhibitors, MEL not only has similar or better anti-inflammatory effects, but also has fewer adverse gastrointestinal effects [10]. MEL acts primarily through the selective inhibition of COX-2. COX-2 is one of the two isoforms of cyclooxygenase and converts arachidonic acid into prostaglandins, which leads to inflammatory reactions and tissue damage [11]. The content of COX-2 is extremely low in normal tissues, and its expression level can be significantly elevated when cells are stimulated by cytokines at the site of inflammation [12].

BMECs acts as sentinels of the mammary gland and have a quick reaction capacity to bacterial infection [13]. After pathogen infection, pattern-recognition receptors located on a wide array of cell types such as BMECs detect pathogen-associated molecular patterns, thereby initiating innate immunity [14]. In this process, NF-κB is promptly activated, thereby giving rise to the generation of cytokines. TNF-α, IL-1, and IL-8 are key pro-inflammatory cytokines inducing acute-phase response, fever, and recruitment of neutrophils [15,16]. In bovine mammary glands, the levels of pro-inflammatory cytokines like TNF-α, IL-1β, and IL-8 increased rapidly within 24 h following *K. pneumoniae* infection [17]. Our previous study also showed that *K. pneumoniae* was able to induce an inflammatory response by activating NF-κB in cultured BMECs [18].

Persistent infection with *K. pneumoniae* leads to ultrastructural damage in BMECs [8]. Mitochondria are often targeted by pathogens because of their role in energy metabolism and antimicrobial innate immunity [19]. In a recent study, it has demonstrated that *K. pneumoniae* infection results in mitochondrial damage and dysfunction in BMECs [20]. Mitochondria are a major source of reactive oxygen species (ROS), and abnormal mitochondrial function induces a large amount of ROS production. When ROS levels exceed the antioxidant capacity of the cells, it causes oxidative stress [21]. Usually, oxidative stress further exacerbates mitochondrial dysfunction, ultimately leading to cell damage and death.

In dairy cows with clinical mastitis, MEL in combination with antibiotics therapy was effective in reducing somatic cell counts and the risk of culling cows as well as improving the bacteriological cure [22,23]. Furthermore, meloxicam has showed promising effects in relieving pain, udder edema, and body temperature in endotoxin-induced clinical mastitis [9]. Although current studies have demonstrated the positive efficacy of MEL in the treatment of clinical mastitis, the direct effects of MEL on BMECs are still not fully elucidated. Further research is needed to determine whether MEL has effects on BMECs in response to *K. pneumoniae* infection.

## 2. Materials and Methods

### 2.1. Antibodies

The following primary antibodies were used: p65 (1:2000, Proteintech, Rosemont, IL, USA), COX-2 (1:1000, Wanleibio, Xi’an, China), p-p65 (1:1000, Cell Signaling Technology, Danvers, MA, USA), β-actin (1:1000, Santa Cruz Biotechnology, Dallas, TX, USA), IκBα (1:1000, Cell Signaling Technology, Danvers, MA, USA), Lamin B1 (1:1000, Affinity, Shanghai, China), p-IκBα (1:1000, Cell Signaling Technology, Danvers, MA, USA), Nrf2 (1:1000, Proteintech, Rosemont, IL, USA), HO-1 (1:2000, Abcam, Cambridge, UK), NQO-1 (1:10,000, Abcam, Cambridge, UK), and Keap1 (1:2000, Proteintech, Rosemont, IL, USA).

### 2.2. Culture of BMEC

Holstein dairy cows were provided by the farm of Yangzhou University. The experimental protocol was approved by the Animal Care and Ethics Committee of Yangzhou University (approval ID: 202205130). Clinically healthy mammary tissues were collected from lactating cows to prepare primary mammary epithelial cells as described by Wang et al. [24]. In brief, try to collect the acini from the mammary glands while avoiding connective tissue. The acini tissue was cut into mince and subsequently subjected to digestion using 0.25% type II collagenase (C6885, Sigma-Aldrich, St. Louis, MO, USA) at a temperature of 37 °C for a duration of 2 h. The digested tissue suspension was filtered sequentially through 20- and 80-mesh filters. Then, the tissue suspension was centrifuged and cultured with DMEM/F12 (D8900, Sigma-Aldrich, St. Louis, MO, USA) containing 10% fetal bovine serum (FBS, A3161002CC, Gibco, Newcastle, Australia), 2 mmol/L L-glutamine, 100 U/mL penicillin, and 100 ug/mL streptomycin in an incubator at 37 °C and 5% CO_2_. Purified epithelial cells could be obtained after two passages based on the differential sensitivity of epithelial cells and stromal cells to trypsin.

### 2.3. Preparation of K. pneumoniae

The *K. pneumoniae* strain utilized in the present study was obtained from a milk sample of bovine mastitis [25]. Our previous study showed that this *K. pneumoniae* isolate was able to induce inflammatory response and inhibit autophagy in cultured immortalized BMECs [18]. *K. pneumoniae* were cultured in liquid Luria–Bertani medium until the logarithmic growth phase at 37 °C. Bacterial precipitates were collected through centrifugation at 3000 rpm for a duration of 5 min, and then the PBS was used to resuspend bacterial precipitates. The above operation was repeated three times. Finally, the *K. pneumoniae* isolate was diluted with DMEM/F12.

### 2.4. Cell Viability Analysis

BMECs were seeded into 96-well plates at a density of 1 × 10^3^ per well and cultured in an incubator at 37 °C with 5% CO_2_ until the cell density reached 80% confluence. The cells were washed three times with PBS and then incubated with different concentrations of MEL (M3935, Sigma-Aldrich, St. Louis, MO, USA) for 12 h. After washing with PBS, the cells were cultured with CCK-8 (A311-01, Vazyme, Nanjing, China) working solution for 2 h according to the instructions. Then, the absorbance was detected at a wavelength of 450 nm.

### 2.5. Cell Infection and MEL Treatment

The ratio of *K. pneumoniae* to cells was defined as the multiplicity of infection (MOI). When the cells reached 80% confluence in a 6-well plate, the medium was changed to DMEM/F12 supplemented with 5% FBS but no antibiotics. Cells were first infected with *K. pneumoniae* at a MOI of 10 for 30 min, and then MEL was added to the cultures with a final concentration of 0.5 or 5 μΜ. After cells were treated with *K. pneumoniae* and MEL together for another 1 h or 3 h, the cells were collected for subsequent experiment. To investigate the effects of MEL on inflammatory response and oxidative stress induced by *K. pneumoniae* in BMECs, five groups were assigned: a negative control group (medium only), MEL control group (medium only containing 5 μM MEL), *K. pneumoniae*-infected group (cells were infected with *K. pneumoniae* as described above), and two *K. pneumoniae* + MEL (0.5 or 5 μM) groups (cells were treated with *K. pneumoniae* and MEL together for 1 h or 3 h).

### 2.6. Western Blot Analysis

Total protein was extracted from BMECs on ice with RIPA buffer containing a mixture of protease and phosphatase inhibitors. The nuclear protein was extracted from BMECs according to the method in the instructions (P0027, Beyotime, Haimen, China). Protein concentrations were detected using a BCA kit (P0010, Beyotime, Haimen, China) and adjusted for the same concentration. Proteins were separated by 10% sodium dodecyl sulfate-polyacrylamide gel electrophoresis and then transferred to a polyvinylidene difluoride membrane. The membranes were blocked through immersing in 5% skimmed milk for 1 h at room temperature and incubated with specific primary antibodies at 4 °C overnight. Subsequently, the membranes were incubated with HRP-conjugated goat anti-rabbit (#7074, Cell Signaling Technology, Danvers, MA, USA) or goat anti-mouse (Lot. 366, Medical & Biological Laboratories, Tokyo, Japan) secondary antibody at room temperature for 1 h. Finally, the chemiluminescence ECL assay kit (E412-01, Vazyme, Nanjing, China) was used to detect immunoreactive protein bands. The gray scale of protein bands was quantified using Image J software (Image J 1.54).

### 2.7. RNA Extraction and Real-Time Quantitative PCR

Total RNA was extracted from BMECs using TRIzol reagent (R401-01, Vazyme, Nanjing, China) and quantified by Nanodrop-2000 spectrophotometer. One microgram of total RNA was transcribed into cDNA using reverse transcription kit (R123-01, Vazyme, Nanjing, China). PCR was performed using SYBR qPCR Master Mix (Q311-02/03, Vazyme, Nanjing, China) according to the manufacturer’s instructions on a PCR system. The primer sequences are shown in Table 1. The relative expression of IL-1β, IL-6, IL-8, TNF-α, and COX-2 were normalized to β-actin and calculated using the 2^−ΔΔct^ method as described previously [26].

### 2.8. Detection of Bacterial Load

BMECs were seeded into 24-well plates to ascertain that they were at least 90% confluent. After infection with *K. pneumoniae* for 3 h in the presence of MEL, the cells were washed three times with PBS and then lysed with 0.3% Triton X-100 for 15 min. The lysate was collected and gradient diluted. The appropriate dilutions were inoculated onto LB agar plates, and the numbers of colonies were counted after incubation at 37 °C for 24 h. To investigate the effects of MEL on *K. pneumoniae* load in BMECs, three groups were assigned: *K. pneumoniae*-infected group (cells were infected with *K. pneumoniae* as described above) and *K. pneumoniae* + MEL (0.5 or 5 μM) groups (cells were treated with *K. pneumoniae* and MEL together for 3 h).

### 2.9. Detection of Intracellular ROS

The ROS level was detected using a fluorescent probe DCFH-DA (S0033S, Beyotime, Haimen, China) after cells were treated with *K. pneumoniae* and MEL together for 3 h. The cells were collected and incubated with DCFH-DA (10 μM) for 20 min at 37 °C. Then, the fluorescence of DCF that represents ROS levels was detected using a flow cytometer at a wavelength of 488 nm.

### 2.10. Detection of SOD, CAT, T-AOC, and Malondialdehyde (MDA)

The cells were collected from 6-well plates after infection with *K. pneumoniae* for 3 h in the presence of MEL. Cell lysate was obtained though sonication on ice followed by centrifugation at 4 °C. The activities of SOD and CAT and the levels of T-AOC and MDA were detected using commercial kits from Nanjing Jiancheng Bioengineering Institute (Nanjing, China).

### 2.11. Immunofluorescence

Cells were seeded on cell slides and treated with *K. pneumoniae* and/or MEL in a 24-well plate. The cells were fixed with pre-cooled 4% paraformaldehyde for 15 min, and then underwent a permeabilization treatment with 0.3% Triton×100 at room temperature for 15 min. The permeabilized cells were blocked with 5% BSA at 37 °C for 1 h. To observe the nuclear accumulation of p65 protein, cells were incubated with anti-p65 (dilution ratio of 1:200) at 4 °C for 12 h. Subsequently, fluorescence-conjugated secondary antibody was added to the well and the 24-well plate was placed at 37 °C for 1 h in the dark. Ultimately, cell nuclei were stained with DAPI. The images were inspected and collected by means of a fluorescence microscope.

### 2.12. Statistics

All assays were independently replicated at least three times to obtain experimental data. Data were shown as the means ± standard error of means. Statistical analysis, including one-way ANOVA and the least significant differences test, were carried out through SPSS 21.0 software (IBM, New York, NY, USA). A *p* value less than 0.05 or 0.01 was considered significant between groups.

## 3. Results

### 3.1. Cell Viability and the Concentration of MEL

The effect of MEL on the BMECs’ viability was evaluated by CCK-8 assay. As shown in Figure 1, MEL (0.5 and 5 μM) had no significant effect on the BMECs’ viability compared with the control group (*p* > 0.05). There was no significant difference between the different concentrations (0.5, 5, 10, 20, and 40 μM). Subcutaneous injection of MEL at 0.5 mg/kg body weight is a commonly used dose for adjunctive treatment of clinical mastitis [9,22,27]. The maximum plasma concentration of MEL was 3.47 μM and 4.78 μM after subcutaneous and oral (0.5 mg/kg body weight) administration, respectively, and the plasma concentration of MEL after 72 h was 0.28 μM and 1.42 μM, respectively [28]. Therefore, the concentrations of 0.5 and 5 μM were selected to explore the effects of MEL on oxidative stress and inflammatory response in BMECs.

### 3.2. Bacterial Load

Bacterial load is an important part of bacterial pathogenesis. Therefore, the effect of MEL on the bacterial load was examined in *K. pneumoniae*-infected BMECs. The results indicated that MEL had no effect on *K. pneumoniae* load in BMECs (*p* > 0.05) (Figure 2).

### 3.3. MEL Inhibits the Production of COX-2

MEL is considered a preferred COX-2 inhibitor. First, the effects of MEL on COX-2 production in *K. pneumoniae*-infected BMECs were examined by qPCR and Western blot analysis. As depicted in Figure 3, *K. pneumoniae* infection increased the mRNA and protein expression levels of COX-2 (*p <* 0.01) compared with the control group. MEL (5 μM) suppressed the mRNA expression levels of COX-2 induced by *K. pneumoniae* (*p <* 0.05) compared with the group infected with *K. pneumoniae* alone. The increased protein level of COX-2 induced by *K. pneumoniae* was also inhibited by MEL treatment (*p* < 0.05 or *p* < 0.01). Moreover, MEL alone can suppress the protein level of COX-2 compared with the control group (*p <* 0.01), but had no effect on the mRNA expression level of COX-2 (Figure 3B,C) in uninfected BMECs. Together, these data demonstrate that MEL dampened the elevated production of COX-2 induced by *K. pneumoniae* in BMECs.

### 3.4. Inflammatory Gene Expression

The effects of MEL on the pro-inflammatory gene’s expression were detected. After *K. pneumoniae* infection, the mRNA expression levels of IL-1β, IL-6, IL-8, and TNF-α were significantly increased (*p* < 0.01) compared with the control group (Figure 4A–D). MEL (0.5 and 5 μM) inhibited the elevation expression of IL-1β and IL-6 (*p* < 0.05 or *p* < 0.01) induced by *K. pneumoniae* compared with the group infected with *K. pneumoniae* alone. In the same context, only 5 μM of MEL significantly inhibited the elevation expression of TNF-α (*p* < 0.05). However, MEL significantly increased the mRNA expression levels of IL-8 compared with the group infected with *K. pneumoniae* alone. These data suggest that the anti-inflammatory effects of MEL on mastitis caused by *K. pneumoniae* is related to inhibition of the production of IL-1β, IL-6, and TNF-α in BMECs.

### 3.5. NF-κB Signaling Pathway Activation

Considering the important role of NF-κB in the inflammatory process, the present study further investigated the effect of MEL on the NF-κB signaling pathway. The activation of the NF-κB signaling pathway was determined by Western blot and immunofluorescence analysis. As illustrated in Figure 5A–C, the phosphorylation levels of p65 and IκBα proteins were significantly increased in *K. pneumoniae*-infected cells compared with the control group (*p* < 0.01). MEL treatment was able to significantly reduce the phosphorylation levels of p65 (*p* < 0.05 or *p* < 0.01) compared with the group infected with *K. pneumoniae* only. In the same context, 5 μM of MEL treatment inhibited the increased phosphorylation of IκBα proteins induced by *K. pneumoniae* (*p* < 0.01). The results of immunofluorescence also showed that *K. pneumoniae* was able to induce the nuclear accumulation of the p65 protein. Compared with the group infected with *K. pneumoniae* alone, the nuclear accumulation of the p65 protein induced by *K. pneumoniae* infection was reduced in the presence of MEL (Figure 6). These results indicate that MEL could inhibit the activation of the NF-κB signaling pathway.

### 3.6. Intracellular ROS and MDA Content

The oxidative milieu of cells was investigated by measuring ROS and MDA. The changes in ROS and MDA in the cells are shown in Figure 7A–C. *K. pneumoniae* infection significantly increases the levels of ROS and MDA (*p* < 0.01) compared with the control group. MEL was able to significantly reduce the increased levels of ROS and MDA induced by *K. pneumoniae* (*p* < 0.01).

### 3.7. T-AOC and Activities of SOD and CAT

The T-AOC, SOD, and CAT were determined to investigate the effects of MEL on the antioxidant capacity of BMECs. The results showed that *K. pneumoniae* reduced cellular SOD, CAT activity, and T-AOC levels (*p* < 0.01) compared with the control group (Figure 8A–C). MEL was able to elevate the activation of SOD and CAT (*p* < 0.01), and 5 μM of MEL improved the level of T-AOC (*p* < 0.05) compared with the *K. pneumoniae* group. These data prove that MEL can improve the antioxidant capacity and thus reduce the oxidative stress state in *K. pneumoniae*-infected cells.

### 3.8. Nrf2 Signaling Pathway Activation

Nrf2 is best known for its role in activating the antioxidant response and protecting cells from damage caused by ROS and electrophiles. Thus, the role of the Nrf2 signaling pathway on the antioxidant protective effect of MEL was explored in the *K. pneumoniae*-infected cells. The expression changes in key proteins in the Nrf2 signaling pathway are shown in Figure 9A–F. Keap1 is a key negative regulator protein of Nrf2, and the protein expression levels of Keap1 and Nrf2 were significantly reduced after *K. pneumoniae* infection (*p* < 0.05 or *p* < 0.01) compared with the control group. Consistently, the Nrf2 nuclear accumulation was increased in the infected cells. However, the antioxidant proteins (HO-1 and NQO1) regulated by the Nrf2 signaling pathway were significantly reduced in the infected cells (*p* < 0.05) compared with the control group, which indicates that *K. pneumoniae* induced the activation of the Nrf2 signaling pathway but may inhibit the regulatory role of Nrf2. MEL increased the Nrf2 nuclear accumulation and the protein expression levels of Nrf2, Keap1, HO-1, and NQO1 (*p* < 0.05 or *p* < 0.01) compared with the group infected with *K. pneumoniae* only. Collectively, these results suggest that MEL can activate the Nrf2 signaling pathway and promote the expression of downstream antioxidant proteins.

## 4. Discussion

Bovine mastitis is recognized worldwide as a disease that poses a great threat to dairy farming, as it leads to huge economic losses and animal welfare issues. *K. pneumoniae* is receiving more attention in herds. This pathogen tends to cause more severe mammary lesions and milk loss than *E*. *coli* [3]. MEL has been approved for supportive therapy for mastitis, in combination with antibiotics, to relieve pain and inflammation. However, its specific effects on BMECs in response to specific pathogens are unknown. A previous study found that MEL at a concentration of 1.5 mg/mL (4268.64 μM) can inhibit the immune response of BMECs in response to LPS and LTA (from *E. coli* and *Staphylococcus aureus*) by preventing increased mRNA expression, such as TNF-α, IL-8, and serum amyloid A [29]. However, the concentration of MEL (4268.64 μM) used in the previous study far exceeded the maximum plasma concentration (4.78 μM) in dairy cows that received the recommended therapeutic dose (0.5 mg/kg body weight). Therefore, plasma maintenance concentrations of MEL (0.5 and 5 μM) were selected to clarify its effects on the inflammatory response and oxidative stress in BMECs infected with *K. pneumoniae*. This study demonstrated that MEL at plasma maintenance concentrations exerted no influence on the viability of uninfected BMECs and had also no impact on bacterial load in BMECs. At these concentrations, MEL can inhibit the inflammatory response and increase the antioxidant capacity of BMECs in response to *K. pneumoniae* infection.

BMECs play a key role in mammary glands against bacterial infection. Cultured BMECs are commonly used to mirror the pathogenicity of different pathogens [30,31,32]. After pathogen infection, BMECs can produce enzymes, cytokines, and chemokines, which are involved in the inflammatory response. COX-2 is an inducible enzyme that contributes to inflammation by catalyzing the production of prostaglandins [33]. A high expression of COX-2 has been found in LPS-stimulated BMECs [34]. The expression of COX-2 is primarily regulated at the transcriptional level by various transcription factors, including NF-κB, AP-1, and STAT-3 [35,36]. In the present study, MEL inhibited the protein level of COX-2 in infected and uninfected cells. Moreover, MEL inhibited the elevated mRNA expression of COX-2 induced by *K. pneumoniae* but had no effect on uninfected cells. These data demonstrate that MEL can inhibit the production of COX-2 at the transcriptional level during inflammation in addition to inhibiting COX-2 activity. IL-1β, TNF-α, and IL-6 are important cytokines that are involved in the mammary immune response. The increased mRNA expression levels of IL-1β, TNF-α, and IL-6 were found in *K. pneumoniae*-infected BMECs [8]. In the present study, MEL inhibited the increased mRNA expression of IL-1β, TNF-α, and IL-6 induced by *K. pneumoniae*. IL-8, as a chemokine, is produced by various types of cells including macrophages, epithelial cells, and fibroblasts. Under inflammation conditions, IL-8 is released and able to recruit and activate neutrophils to the sites of inflammation [37]. It was found that dairy cows with higher levels of IL-8 in whey were more capable of limiting *E. coli* growth in the mammary glands [38]. In the current study, the mRNA abundance of IL-8 was upregulated in BMECs after *K. pneumoniae* infection. Interestingly, the addition of MEL further increased the mRNA abundance of IL-8. This result may partly explain why MEL treatment improves the bacteriological cure in mastitis [23], which may be related to its enhancement of the production of IL-8 and recruitment of more neutrophils to the infected mammary gland. However, the underlying molecular mechanisms by which MEL promotes the mRNA expression of IL-8 require further study.

NF-κB is a typical signaling pathway that regulates the pro-inflammatory response to infection [39]. The genes’ expression of cytokines, chemokines, and enzymes that produce inflammatory mediators are largely based on the activation of canonical NF-κB in infected BMECs [40,41,42]. The activation of the canonical NF-κB pathway requires IκB kinase (IKK) subunits and the nuclear translocation of NF-κB p65 [43]. In normal cells, the p65 is maintained in the cytoplasm due to binding the inhibitory protein IκBα. IKKβ regulates the activation of the NF-κB pathway by phosphorylating IκBα. Phosphorylated IκBα is subsequently ubiquitinated and degraded, resulting in the translocation of p65 from the cytoplasm to the nucleus. The p65 enters the nucleus to promote the transcription of genes encoding cytokines, chemokines, and enzymes [44]. Our previous study showed that *K. pneumoniae* was able to activate the NF-κB signaling pathway in BMECs by phosphorylating the IκBα and p65 [25], which was again demonstrated in this study. The activation of the NF-κB signaling pathway induced by *K. pneumoniae* can be inhibited by MEL. Collectively, these results indicate that NF-κB inhibition is one of the protective mechanisms through which MEL protects cells from inflammatory damage.

Oxidative stress and inflammation have an interdependent relationship and are simultaneously present in infection conditions [45,46]. Since MEL can inhibit the inflammatory response, it may also affect oxidative stress in *K. pneumoniae*-infected cells. The extent of oxidative stress depends on the balance between ROS production and antioxidant defenses. A deficiency in antioxidants or an excessive production of ROS can lead to oxidative stress and cell injury. ROS are an important cause of oxidative stress. They attack various biomolecules, including proteins, DNA, and lipids. Among them, lipids are the most vulnerable to ROS, which is known as lipid peroxidation [47]. MDA is the product of lipid peroxidation [48]. SOD and the catalase CAT are important enzymatic antioxidants that reduce oxidative damage [49]. In this study, *K. pneumoniae* infection induced oxidative stress, manifested as increased ROS and MDA levels, and reduced the activity of SOD and CAT and the level of T-AOC. As expected, oxidative stress that occurred in *K. pneumoniae*-infected cells can be alleviated in the presence of MEL.

The Nrf2 signaling pathway is a complex network that orchestrates cellular responses to oxidative stress. Nrf2 activity is primary regulated by redox-sensitive cysteine residues within the Keap1 protein. Excessive ROS react with these cysteines, leading to the deactivation of Keap1. Once Keap1 is deactivated, Nrf2 is released from its interaction with Keap1 and can translocate to the nucleus. Then, Nrf2 binds to antioxidant response elements (AREs) that are present in genes encoding antioxidant enzymes like NQO-1, HO-1, SOD, and CAT, which leading to their transcription [50]. Disruption of redox signaling contributes to imbalances between oxidants and antioxidants [51]. In this study, *K. pneumoniae* infection induced the Nrf2 nuclear accumulation; however, it subsequently inhibited the expression of Nrf2 downstream proteins. We hypothesized that Nrf2 can translocate to the nucleus, but the function of regulating the transcription of downstream genes is inhibited in *K. pneumoniae*-infected cells. Our results showed that MEL could improve the antioxidant capacity by promoting Nrf2 translocation into the nucleus, elevating the expression of HO-1, NQO1, and other antioxidant enzymes.

## 5. Conclusions

Plasma maintenance concentrations of MEL (0.5 and 5 μM) were able to reduce the inflammatory responses and oxidative stress in *K. pneumoniae*-infected BMECs. The mechanisms by which MEL attenuates inflammatory responses and oxidative stress is at least partially related to its inhibition of the NF-κB signaling pathway and activation of the Nrf2 signaling pathway. However, the mechanism by which MEL promotes the mRNA expression of IL-8 requires further study. Furthermore, the function of Nrf2 regulating the downstream genes transcription was disturbed in *K. pneumoniae*-infected cells, which should gain more attention.

## Figures and Tables

**Figure 1 vetsci-11-00607-f001:**
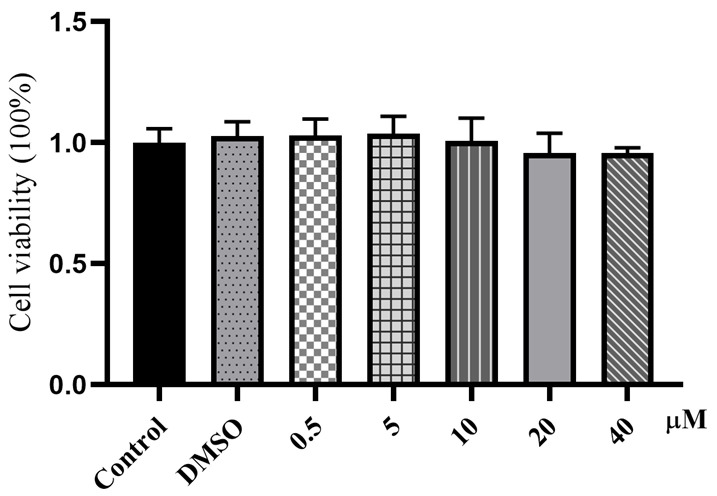
The effects of MEL on cell viability. BMECs were treated with different concentrations (0, 0.5, 5, 10, 20, and 40 μΜ) of MEL for 12 h. The cell viability was evaluated with the CCK-8 method.

**Figure 2 vetsci-11-00607-f002:**
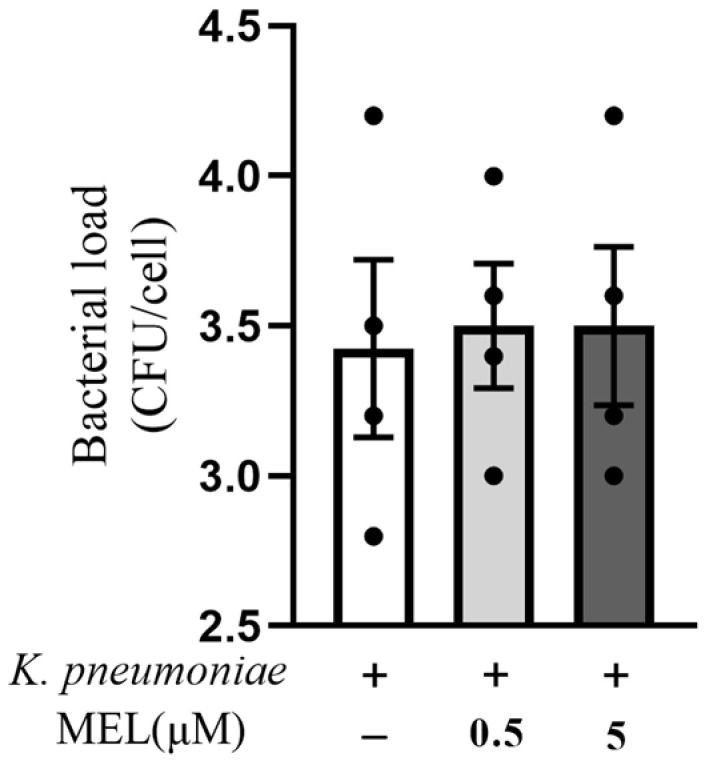
The effects of MEL on bacterial load. BMECs were infected with *K. pneumoniae* for 3 h in the presence or absence of MEL. Each experiment was repeated 4 times.

**Figure 3 vetsci-11-00607-f003:**
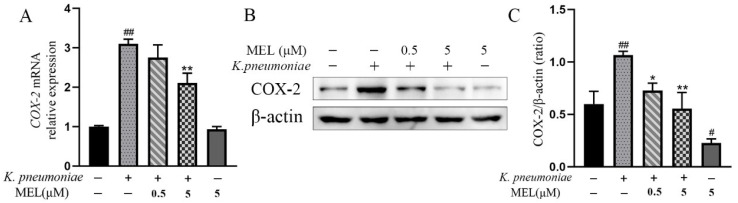
The effects of MEL on the mRNA expression (**A**) and protein levels (**B**,**C**) of COX-2 in BMECs. BMECs were infected with *K. pneumoniae* for 3 h in the presence or absence of MEL. *# p* < 0.05 and ## *p* < 0.01 compared with the control group. * *p* < 0.05 and ** *p* < 0.01 compared with the group infected with *K. pneumoniae* alone (see Appendix A).

**Figure 4 vetsci-11-00607-f004:**
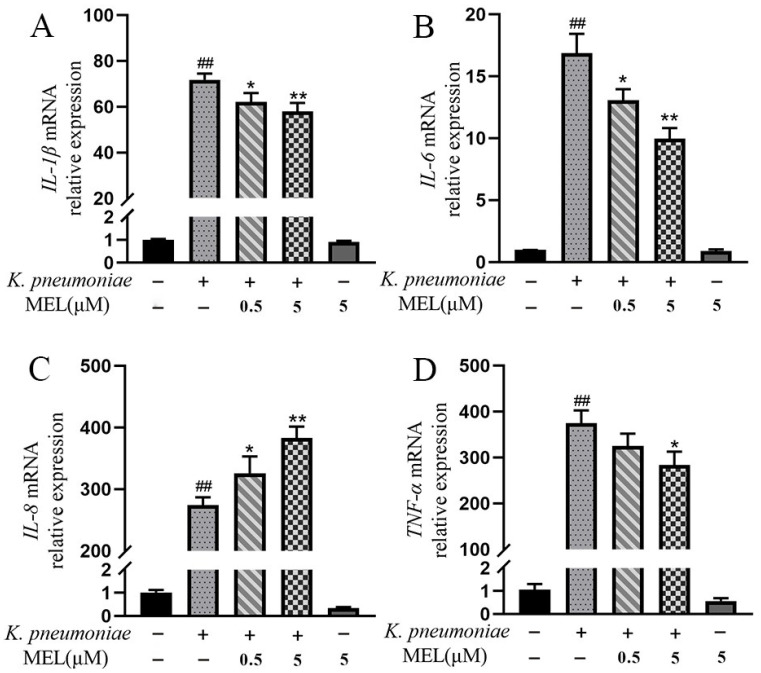
The effects of MEL on the mRNA expression of IL-1β (**A**), IL-6 (**B**), IL-8 (**C**), and TNF-α (**D**) in BMECs. BMECs were infected with *K. pneumoniae* for 3 h in the presence or absence of MEL. ## *p* < 0.01 compared with the control group. * *p* < 0.05 and ** *p* < 0.01 compared with the group infected with *K. pneumoniae* alone.

**Figure 5 vetsci-11-00607-f005:**
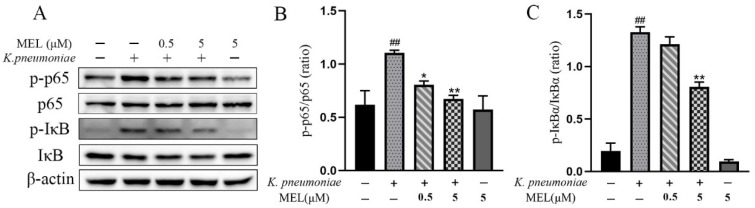
The effects of MEL on the NF-κB signaling pathway. BMECs were infected with *K. pneumoniae* for 1 h in the presence or absence of MEL. (**A**) The protein expression of p-p65, p65, p-IκBα, and IκBα in BMECs (see Appendix A). (**B**) Changes in the phosphorylation level of p65. (**C**) Changes in the phosphorylation level of IκBα. ## *p* < 0.01 compared with the control group. * *p* < 0.05 and ** *p* < 0.01 compared with the group infected with *K. pneumoniae* alone.

**Figure 6 vetsci-11-00607-f006:**
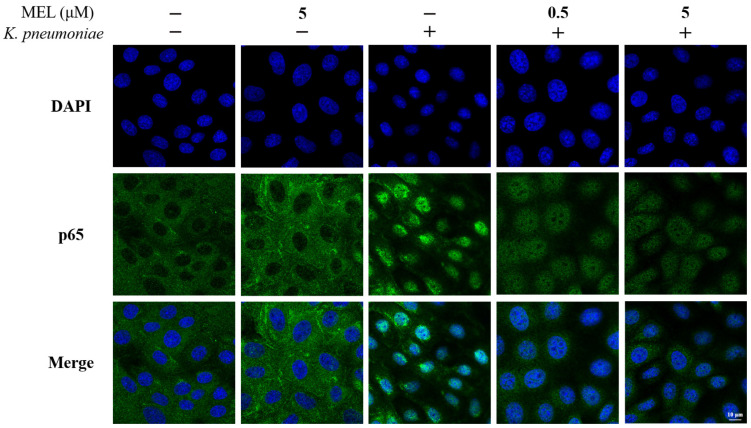
The effect of MEL on the nuclear accumulation of the p65 protein in BMECs. BMECs were infected with *K. pneumoniae* for 1 h in the presence or absence of MEL.

**Figure 7 vetsci-11-00607-f007:**
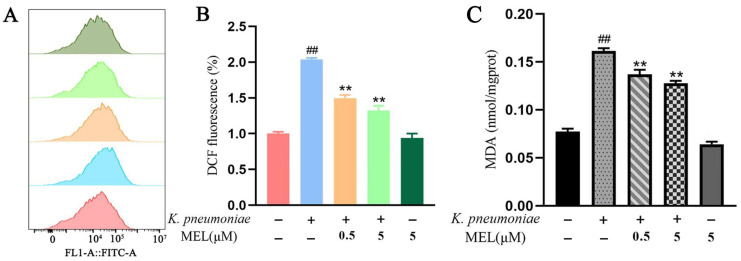
The effects of MEL on the oxidative state of BMECs. (**A**,**B**) Changes in the level of ROS. (**C**) Changes in the level of MDA. BMECs were infected with *K. pneumoniae* for 3 h in the presence or absence of MEL. ## *p* < 0.01 compared with the control group. ** *p* < 0.01 compared with the group infected with *K. pneumoniae* alone.

**Figure 8 vetsci-11-00607-f008:**
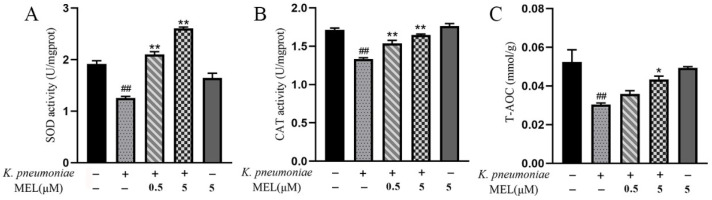
The effects of MEL on the antioxidant capacity of BMECs. BMECs were infected with *K. pneumoniae* for 3 h in the presence or absence of MEL. The activity of SOD (**A**) and CAT (**B**) and the level of T-AOC (**C**) were detected using commercial kits. ## *p* < 0.01 compared with the control group. * *p* < 0.05 and ** *p* < 0.01 compared with the group infected with *K. pneumoniae* alone.

**Figure 9 vetsci-11-00607-f009:**
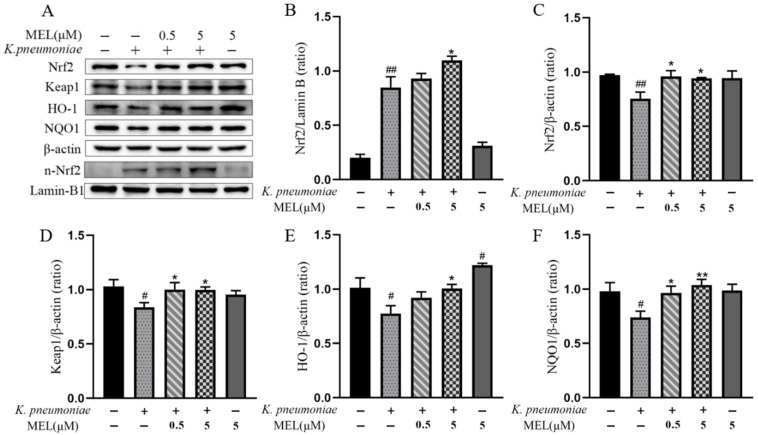
The effects of MEL on the Nrf2 signaling pathway in BMECs. (**A**) The key proteins expressed in the Nrf2 signaling pathway were detected by Western blot (see Appendix A). (**B**) Changes in Nrf2 nuclear accumulation. Nuclear protein was extracted from cells. The protein expression levels of Nrf2 (**C**), Keap1 (**D**), HO-1 (**E**), and NQO1 (**F**) were detected using total protein. BMECs were infected with *K. pneumoniae* for 3 h in the presence or absence of MEL. *# p* < 0.05 and ## *p* < 0.01 compared with the control group. * *p* < 0.05 and ** *p* < 0.01 compared with the group infected with *K. pneumoniae* alone.

**Table 1 vetsci-11-00607-t001:** Primer sequences for PCR (F = forward, R = reverse).

Gene	Primer Sequences (5′-3′)	Accession Number
*β-actin*	F: CATCACCATCGGCAATGAGC R: AGCACCGTGTTGGCGTAGAG	NM_173979.3
*IL-1β*	F: GCTATGAGCCACTTCGTGAGGAC R: GATTGAGGGCGTCGTTCAGGAT	NM_174093.1
*IL-6*	F: TGATGACTTCTGCTTTCCCTACCC R: ATCTTTGCGTTCTTTACCCACTCG	NM_173923.2
*IL-8*	F: ATGACTTCCAAGCTGGCTGTT R: GGTTTAGGCAGACCTCGTTTC	NM_173925.2
*TNF-α*	F: GCTCTCTCTCACATACCCTGC R: ATCCCGGATCATGCTTTTGGT	NM_173966.3
*COX-2*	F: TCCTGAAACCCACTCCCAACA R: TGGGCAGTCATCAGGCACAG	AF_031698.1

## Data Availability

The data presented in this study are available in the article.

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
