# Peer review of "The Effect of Meloxicam on Inflammatory Response and Oxidative Stress Induced by Klebsiella pneumoniae in Bovine Mammary Epithelial Cells"

_vetsci, 2024, doi:10.3390/vetsci11120607_

Round 1
Reviewer 1 Report
Comments and Suggestions for Authors
Review – The Effect of Meloxicam on Inflammatory Response and Oxidative Stress induced by Klebsiella pneumoniae in Bovine Mammary Epithelial Cells-
The article describes a study attempting to determine the efficacy of meloxicam (MEL) on the immune response and oxidative stress of mammary gland epithelial cells with K. pneumoniae infections in vitro. The procedure is mostly carefully and clearly described. An influence of MEL on cell viability and bacterial load was not determined in the dose range investigated. However, an influence on various cytokines and on oxidative stress was found, which led to the conclusion that MEL in usual plasma concentrations can protect mammary gland epithelial cells from inflammatory and oxidative damage
In my opinion, the journal for the article is not well chosen. Even if the title sounds practice-oriented, it is more of a basic paper with little direct practical relevance.
Abstract:
The study design should be better described in the abstract. Abbreviations should be explained in the abstract.
Introduction:
44 apoptosis? Cell death/ necrosis
Material and Methods:
Missing: Overall conclusion of which experiments where done (controls / which assignments where compared to each other)
Results:
3.1 Infected or uninfected BMECs? Was there a comparison between infected and uninfected BMECs?
Discussion:
328 ff was there a comparison between infected and uninfected BMECs?
Why is there no effect on cell viability when there is less oxidative stress and inflammation?
Figure 7A.: label is not readable
74 – 75: When ROS levels exceed the antioxidant capacity of the cells, which it causes oxidative stress.
84 – 85: Further research is needed
195: subcutaneous
335: … which are involved in an inflammatory response.
343: … MEL can inhibited …
345: … that are involved …
369: … by phosphorylating …
379: ROS are an important cause …
381: … which is known …
Author Response
Thank you for your comments concerning our manuscript. Those comments are very helpful for revising and improving our paper and, also the important guiding significance to our researches. We have studied those comments carefully and have made correction which we hope meet with your approval. Please see the attachment.

Reviewer 2 Report
Comments and Suggestions for Authors
This study aimed to demonstrate that MEL at plasma maintenance concentrations could protect BMECs from inflammatory and oxidative damage caused by K. pneumoniae. The quality of the manuscript are good and comments are in the attachment.

Author Response
Thank you for your comments concerning our manuscript. Those comments are very helpful for revising and improving our paper and, also the important guiding significance to our research. Please see the attachment for the details of the manuscript revisions

Reviewer 3 Report
Comments and Suggestions for Authors
Dear authors,
The present document is related to the most important environmental pathogen causing mastitis in dairy cattle. Therefore, the data and information you offer should be clearer and improve the image presentations to see and understand much better. Please, check all comments made in the document.
The supplementary information, review all figures, and choose only the best images that represent all work made.

Author Response
Thank you for your comments concerning our manuscript. Those comments are very helpful for revising and improving our paper and, also the important guiding significance to our research. We have studied those comments carefully and have made correction which we hope meet with your approval. Please see the attachment for the details of the manuscript revisions.

Round 2
Reviewer 1 Report
Comments and Suggestions for Authors
Thank you for your changes. I think, that the paper is more or less ready to publish. Some typos should be fixed:
l. 24 delete "the"
l. 60 cell instead of cells
l. 205 All instead of ALL